# Polioencephalomalacia in Buffaloes in the Amazon Biome

**DOI:** 10.3390/ani13193131

**Published:** 2023-10-07

**Authors:** José Diomedes Barbosa, Fernanda Monik Silva Martins, Carlos Eduardo da Silva Ferreira Filho, Camila Cordeiro Barbosa, Eliel Valentim Vieira, Ruama Paixão de Lima Silva, Cinthia Távora de Albuquerque Lopes, Natália da Silva e Silva Silveira, Carlos Magno Chaves Oliveira, Marilene Farias Brito, Felipe Masiero Salvarani

**Affiliations:** 1Instituto de Medicina Veterinária, Universidade Federal do Pará, Castanhal 68740-970, PA, Brazil; diomedes@ufpa.br (J.D.B.); nanda.monik08@gmail.com (F.M.S.M.); carlos.filho@ananindeua.ufpa.br (C.E.d.S.F.F.); camilabarbosamedvet@gmail.com (C.C.B.); vieiraeliel2002@gmail.com (E.V.V.); ruama-paixao@hotmail.com (R.P.d.L.S.); cinthia@ufpa.br (C.T.d.A.L.); nataliasilvasilveira1@gmail.com (N.d.S.e.S.S.); cmagno@ufpa.br (C.M.C.O.); 2Departamento de Epidemiologia e Saúde Pública (DESP), Instituto de Veterinária (IV), Universidade Federal Rural do Rio de Janeiro (UFRRJ), Seropédica 23890-000, RJ, Brazil; mfariasbrito@uol.com.br

**Keywords:** neurological disease, neurological clinical signs, thiamine metabolism, diagnostic, epidemiological and clinicopathological aspects

## Abstract

**Simple Summary:**

The buffalo industry has great importance in the state of Pará, within the Amazon biome, contributing to society in economic and cultural ways. For this reason, it is necessary to understand the disorders that affect this species, such as polioencephalomalacia (PEM), which has not been reported in the Brazilian literature. This is the first study about PEM in buffalo in the Amazon biome. This work describes five disease cases in buffaloes, presenting their clinical, macroscopic and histopathological findings.

**Abstract:**

The epidemiological and clinical pathological aspects of polioencephalomalacia (PEM) in five buffaloes in the Amazon biome are described. Epidemiological data were obtained during the clinical visit in which buffalos were submitted to an examination of the nervous system. The diagnosis of PEM was based on epidemiological, clinical–pathological, and histopathological findings, similar to findings in other ruminants. In the clinical examination, all buffaloes had a body score between 2.5 and 3 (scale from 1 to 5), with decreased alertness, postural changes, marked hypermetria when stimulated to move, total or partial blindness demonstrated by colliding with the corral structures, head pressing and circumduction of the hindquarters when supported on the thoracic limbs, opisthotonos, muscle tremors, convulsions, paddling movements, sialorrhea, decrease in palpebral and pupillary reflexes, and eyeball rotation placing the pupillary slit in a vertical position. At necropsy, mild lesions characterized by edema and flattening of the cerebral convolutions were observed. A histopathological examination showed laminar necrosis of the cerebral cortex. The cause of PEM in the studied buffaloes was not established, which indicates the need for further studies to elucidate this disease in the species, particularly in the Amazon region.

## 1. Introduction

Among the diseases that affect ruminants, central nervous system (CNS) disorders cause numerous losses, especially rabies and botulism [1]. However, other conditions, such as bovine herpesvirus type 5 and polioencephalomalacia (PEM), also deserve mention. In this context, PEM is a descriptive term that means necrosis with softening (malacia) of the gray matter (polio) of the brain [2,3]. The use of this term can be confusing as it can be used to designate the laminar necrosis lesion of the cerebral cortex, which is seen in sodium chloride intoxication, sulfur and lead poisoning, and bovine herpesvirus type 5 encephalitis, as well as specific neurological diseases in ruminants associated with disturbances in thiamine (vitamin B1) metabolism [3,4].

PEM is a condition that affects animal CNSs, causing neurological signs associated with different brain structures. In the early stages, flaccidity is common, with transient periods of spasticity and intermittent chronic seizures and muscle tremors, especially of the head, and intermittent to permanent opisthotonos. In severe cases with a longer evolution of PEM, the animals become prostrate, with ear, eyelid, and facial spasms; empty intermittent chewing; teeth grinding; sialorrhea; hyperexcitability; and aggressiveness [5,6].

At necropsy, there is encephalic edema, pallor, and softening of the cerebral hemispheres and flattening of the cerebral gyri, with herniation of the cerebellum through the foramen magnum and extensive gray matter malacia areas [4,5,6]. The diagnosis of PEM is based on epidemiological, clinical–pathological, and histopathological findings and, in some cases, on the clinical response to thiamine and corticoid treatment [7].

PEM was initially described in cattle and sheep with neurological disorders in the state of Colorado, USA. The etiology was attributed to thiamine deficiency and later to multifactorial characteristics of the disease [2,8]. In Europe, the same lesion pertinent to PEM was called cerebrocortical necrosis [8], and there are even reports of the disease in llamas in Canada [9]. In Brazil, PEM has been described in bovines [10,11,12], ovine [13,14,15], goats [13,14,16], and buffaloes in the state of Mato Grosso, midwest Brazil [17].

The buffalo industry has great importance in the state of Pará, within the Amazon biome, contributing to society in economic and cultural ways. For this reason, it is necessary to understand the disorders that affect this species, such as PEM. Thus, considering the low number of references on PEM in buffaloes, the objective of the present work is to report five PEM cases in buffaloes, four in the state of Pará and one in the state of Amapá, within the Brazilian Amazon biome.

## 2. Materials and Methods

This study comprised five buffaloes (numbered 01 to 05). All were of the Murrah breed, four male and one female, with ages ranging from two months to one year. Two buffaloes (buffalo 01 and 02) were from the municipality of Castanhal, Pará, kept in an extensive breeding system on the *Urochloa brizantha* pasture. Buffaloes 03 and 04 belonged to a property in Cachoeira do Arari, Marajó Island, Pará. Buffalo 05 was from a property in Itaubal, Amapá. All were raised in an extensive system in a flooded native pasture.

Epidemiological data, such as age, sex, race, farm location, and breeding systems, were obtained at the clinical visit. As established by Dirksen et al. [18], animals with suspicion of PEM were submitted to a general and specific clinical examination. Necropsy was performed on the five animals. During the necropsy, the location and intensity of the lesions were evaluated and organ fragments were collected, fixed in 10% buffered formalin, and sent for histopathological examination in the Federal Rural University of Rio de Janeiro. The fragments were submitted to the methodology described by Luna [19]. All animals were negative for rabies and botulism, [20]. This research was authorized by the animal experimentation ethics committee (CEUA) of the Federal University of Pará (UFPA) under protocol number 6261300323 (ID 002208).

## 3. Results

The diagnoses of the five PEM buffaloes were based on epidemiological, clinical–pathological, and histopathological studies—this being the first report of the disease in the Amazon biome. At the clinical examination, all buffaloes had a body score of 2.5–3 (scale from 1 to 5). Buffaloes 03 and 04 were still standing at the clinical examination, and had a noted decreased alertness; postural changes, marked hypermetria when wandering; blindness, demonstrated by colliding with the corral structures; head pressing (Figure 1a); and pelvic limbs circumduction when supported on the thoracic limbs. These two buffaloes evolved to lateral decubitus. Buffaloes 01, 02, and 05 were in lateral decubitus at the clinical visit and presented opisthotonos (Figure 1b), muscle tremors, convulsions, paddling movements, sialorrhea, and decreased palpebral and pupillary reflexes. Additionally, buffaloes 01 and 05 had rotation of the eyeball, placing the pupillary slit in a vertical position (cat pupil) (Figure 1c).

In terms of clinical course, all animals died between three and five days after the clinical signs’ onset. Posteriorly, at necropsy, mild lesions characterized by edema and flattening of the cerebral convolutions were evidenced. In addition to the alterations observed in the brain in buffalo 01 and 05, the contents of the rumen had a pasty appearance, with a “grease appearance” resulting from a mixture of mud and plant material (Figure 2a) and abundant parasites in the rumen (Figure 2b). Histopathological findings showed laminar necrosis of the cerebral cortex (Figure 2c).

## 4. Discussion

Due to the scarce literature on PEM in buffaloes, we considered the studies carried out in cattle, goats, and sheep as references.

The ages of the animals studied ranged from two months to one year. Withoeft et al. [12] diagnosed PEM in calves from 15 days to 1 year. This can be explained by primary thiamine deficiency, as the young cannot produce or ingest it in sufficient quantities [8,21]. Furthermore, according to Dal Mas et al. [7], PEM does not show seasonality, with foci occurring in all months of the year in Mato Grosso do Sul, a fact that also occurs in Australia, the USA, and Great Britain. Therefore, the tropical climate of the Amazon region is not a determining factor for the occurrence of PEM in buffaloes, demonstrating that PEM can also occur in this climate.

Guimarães et al. [17] diagnosed PEM in buffaloes in leguminous pastures, as evidenced by the possibility of PEM emergence in buffaloes bred in different pastures. Moro et al. [22] also reported cases of PEM in cattle with sudden changes in feeding from poor to excellent, and generally, young animals on a high concentrate diet are at high risk [23]—a fact that did not occur in buffaloes in the outbreaks studied.

The clinical signs were of a multifocal nature, involving all regions of the brain (e.g., cerebellum and brainstem), similar to the clinical signs in cattle [10,11,12], sheep [13], and goats [14,16]. In light of this symptomatology, Riet-Correa et al. [5] suggest that the increase in intracranial pressure exerted by edema of the brain structures causes herniation of the cerebellum through the foramen magnum.

The death of buffaloes between the third and fifth day after the appearance of symptoms demonstrates an acute evolution of this disease in buffaloes, consistent with Sampaio et al. [6], who referred to the evolution of PEM from a few hours to four days.

The intensity of the lesions could be due to the acute clinical course since Riet-Correa et al. [5] report that necropsy findings vary according to the severity and duration of the clinical condition. Commensurate with the literature, the lesions are discrete and there is a decrease in brain consistency in animals with an acute clinical course, as observed in this study [3]. Furthermore, due to cerebral edema, caudal displacement (herniation) of the cerebellum through the foramen magnum may occur [3,5]. In chronic cases, in which the animals survive for several days, necrosis becomes more evident and the brain is visibly reduced in size due to the loss of gray matter, with flattening of the convolutions, yellowish gelatinous consistency areas of the cortex, and cavitations [5,6]. Laminar necrosis of the cerebral cortex has also been found in cattle with PEM [10,12,24] and sheep intoxicated with sulfur [13].

The ruminal content of buffalo 01 and 05 was considered an unsuitable environment for bacterial activity. As ruminants depend on ruminal bacteria thiamine synthesis, we assume that in this unsuitable environment, thiamine production was insufficient to meet the animals’ needs, triggering the disease [2,21]. The active form of thiamine is thiamine pyrophosphate, a cofactor of transketolase, which regulates the pentose phosphate pathway responsible for much of the ATP used in nerve cells [8], a fact that explains the triggering of PEM in these animals. In addition, sulfur intoxication cannot be ruled out as it leads to lesions in PEM conditions involving ruminal bacteria, transforming ingested sulfates to the toxic form (sulfides), which affects cellular respiration [12,21].

Regarding the differential diagnosis, it is important to consider diseases that cause laminar necrosis of the cerebral cortex, such as lead poisoning, encephalitis due to bovine herpes virus type 5, and salt intoxication [3]. However, the animals in this study were raised in an extensive system and did not receive mineral supplementation or have access to lead sources, which excludes the possibility of salt or lead poisoning. Additionally, in encephalitis by herpesvirus type 5, common histopathological findings are extensive areas of malacia in the telencephalon cortex, with possible intranuclear inclusion bodies in astrocytes and neurons [25,26,27], which was not observed in the studied animals.

Considering the most frequent diseases, rabies was ruled out by epidemiological data (as the disease only affects young buffalo calves) and by histopathological findings such as the absence of the Negri corpuscle in the CNS tissue and the presence of CNS gray matter necrosis, which is not correlated with rabies [28]. Likewise, botulism was also ruled out by epidemiology, as this intoxication is not common in suckling calves and the typical clinical signs (e.g., flaccid paralysis) were not observed in the studied buffaloes [29].

The lack of PEM studies in buffaloes could be due to several factors: the fact that the prevalence of PEM in buffaloes could be lower, making it less of a priority for research compared to more common diseases; research efforts are often directed towards diseases that have a significant economic impact on the livestock industry; conducting thorough scientific studies requires resources and if the buffalo population with PEM is limited and scattered, it might be challenging to gather enough cases for comprehensive research; PEM affects various ruminant species, including cattle and sheep, and researchers might choose to focus on these more common species; PEM cases in buffaloes are not accurately diagnosed and this lack of awareness could contribute to the scarcity of research interest; and the prevalence of PEM might vary by region due to differences in feed composition, management practices, and environmental conditions.

From the perspective of the disease in buffaloes, it was not possible to establish the cause of PEM in the animals studied. This demonstrates the vital need for descriptive and pathophysiological studies of the disease in buffaloes, mainly in the Amazon region, which has all the epidemiological factors favorable to disease occurrence.

## 5. Conclusions

This is the first study about polioencephalomalacia in *Bubalus bubalis* in the Amazon Biome. The cause of PEM in the studied buffaloes was not established, which indicates the need for further studies to elucidate this disease in the species and how the present study could be expanded (for example, number of buffaloes, various municipalities/farms or pasture types). And finally, to encourage research, it is essential for veterinary and agricultural institutions to recognize the significance of studying PEM in buffaloes and allocate resources accordingly. Collaboration between researchers, veterinarians, and livestock farmers can play a pivotal role in better understanding and managing the disease in buffalo populations.

## Figures and Tables

**Figure 1 animals-13-03131-f001:**
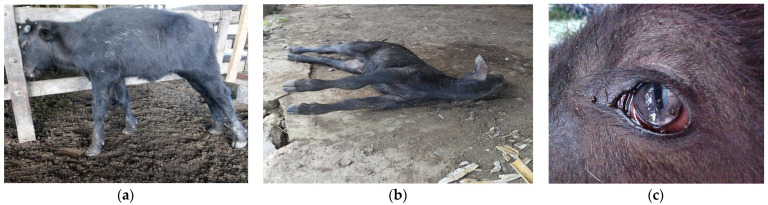
Polioencephalomalacia in young Murrah buffaloes in the Brazilian Amazon biome: (**a**) head pressing against the corral fence, (**b**) animal with limb spasticity and opisthotonos, (**c**) eyeball rotation with the pupillary slit in vertical position.

**Figure 2 animals-13-03131-f002:**
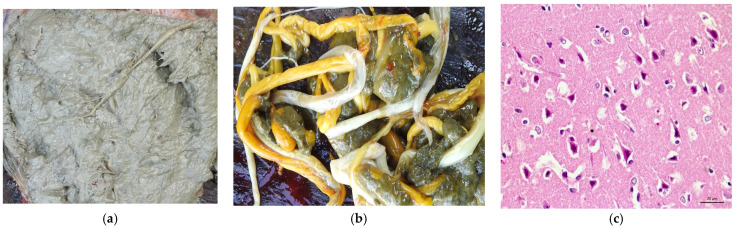
Necropsy and histopathological findings of buffaloes with polioencephalomalacia found in the Brazilian Amazon biome. (**a**) The contents of the rumen had a pasty appearance, with a “grease appearance” resulting from a mixture of mud and plant material; (**b**) parasites in the rumen; (**c**) necrotic (red and contracted) neurons in the cortex (Obj. 40×, H&E).

## Data Availability

Not applicable.

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
