# Peer review of "Polioencephalomalacia in Buffaloes in the Amazon Biome"

_animals, 2023, doi:10.3390/ani13193131_

Round 1
Reviewer 1 Report
The paper refers to an important neurological disease in buffaloes. Especially since they are bred in the Amazon biome. In the ruminant clinic, information about buffaloes is not very frequent. This type of communication is important in South America.
It is well written and clear in objectives, methodology and results.
The bibliographical references are pertinent. But I also suggest something about clinical treatment. The pictures of the sick buffaloes are good, as well the photos of the histopathological examination.
The study was well conducted, bringing news about the buffalo clinic. I just suggest a paragraph about treatment of polioencephalomalacia. In some situations, even with a severe clinical sign, treatment with vitamin B1 (thiamine), can be effective within minutes. We understand it as a clinical emergency. This is one of the rare cases where a “therapeutic diagnosis” can be made. However, if the biochemical problem has already caused encephalic necrosis, the treatment does not work. That´s why we consider a medical emergency.
Why was thiamine treatment not tried? From experience, we always try treatment in these suspected cases. It is an emergency and we have good results in recovery with the treatment (vitamin B1). I think it can be commented on in the description of the clinical case and also in the discussion.

Author Response
Dear reviewer 1,
We appreciate your suggestions and comments about the work. We were honored to have our article recognized by peers during the review process.
Regarding your request to insert a paragraph about the treatment with vitamin B1 (thiamine), unfortunately we were unable to confirm the diagnosis of the real cause of PEM, in addition, the 5 animals studied already had encephalic necrosis, as we observed in the histopathology and by that the treatment does not work. However, we discussed the importance of vitamin B1 and I believe its importance has become clear, as you well put it, "In some situations, even with a severe clinical sign, treatment with vitamin B1 (thiamine), can be effective within minutes. We understand it as a clinical emergency. This is one of the rare cases where a “therapeutic diagnosis” can be made."
The article went through a series of writing and content reviews, requested by the 3 other reviewers besides you, who mainly questioned the issue of scientific writing in English. I hope we have clarified your doubts and met your expectations.
Thank you very much.
Sincerely,
Felipe Masiero Salvarani

Reviewer 2 Report
The manuscript is well described, but, as the authors themselves state, there is no determination of what caused the polioencephalomalacia (PEM). The discussion is built around hypotheses. Therefore, just the unprecedented occurrence of PEM in buffaloes in the Amazon biome is not enough for publication in Animals.
Minor editing of English language required
Author Response
Dear reviewer 2,
We appreciate your suggestions and comments about the work.
With regard to your opinion that although the manuscript is well described only the description of the occurrence of PEM in buffaloes in the Amazon biome is not enough for publication in Animals, we respect, however, it is not the same shared by the editors of the journal and the another 3 reviewers, who recognize the scientific relevance of the work and recommend it for publication.
The article went through a series of writing and content reviews, requested by the 3 other reviewers besides you, who mainly questioned the issue of scientific writing in English. Sorry to disagree with your opinion, but it is part of the review process, in addition to having to defend our work, and I hope that this new version, which continues to be based on hypotheses, can meet your expectations.
Thank you very much!
Sincerely,
Felipe Masiero Salvarani

Reviewer 3 Report
This paper is the first report on the clinical signs and pathological findings in five buffalos with PEM in Amazon biome. This deserves to be in the published literature, and has good photographs of the animals and lesions, but is not publishable as submitted: the english language requires extensive revision (eg"
Buffaloes 03 and 04 were still standing at the clinical examination, and there was noted decreased alertness, postural changes, marked hypermetria when wandering, blindness demonstrated by colliding to the corral structures, head pressing (Figure 1A), and hindlimbs circumduction when supported on the forelimbs"
and
"Posteriorly, at necropsy, mild lesions characterized by edema and flattening of the cerebral convolutions were evidenced" and the paper should be re-written by a native English speaker with knowlede of veterinary anatomy nomenclature (eg animals do not have a 'posterior' aspect and 'hindlimbs' should be 'pel;vic limbs').
Also the arguments as to what might be causing the disease in these five animals is unclear; eg on what analysis was the statement " based on?
The ruminal content found in buffalo 01 and 05 is considered an unsuitable environment for bacterial activity. As ruminants are dependent on the synthesis of thiamine by ruminal bacteria, it leads us to assume that in this ruminal environment found, thiamine production was insufficient to meet the needs of the animal, not ruling out this condition in triggering the disease
I would be happy to review the paper again after it has been re-written.
As above
Author Response
Dear reviewer 3,
We appreciate your suggestions and comments about the work. And we were very honored by the recognition of the scientific merit of the article.
We fully agree that the article needed to undergo an extensive English revision, which was done by MDPI itself, as documented in the annex. Therefore, the article went through a series of writing and content reviews, requested by the 3 other reviewers besides you, who mainly questioned the issue of scientific writing in English. I hope we were able to meet your expectations regarding the new revised article.
Thank you very much!
Sincerely,
Felipe Masiero Salvarani

Reviewer 4 Report
Suggestions and comments are pointed into the file attached below.

Minor editing of English language required
Author Response
Dear Reviewer 4,
We appreciate your suggestions and comments about the work. And we were very honored by the recognition of the scientific merit of the article.
We fully agree that the article needed to undergo an extensive English revision, which was done by MDPI itself, as documented in the annex. Therefore, the article went through a series of writing and content reviews, requested by the 3 other reviewers besides you, who mainly questioned the issue of scientific writing in English. We have made the adjustments you requested and hope that we have been able to meet your expectations regarding the new revised article.
Thanks you very much!
Sincerely,
Felipe Masiero Salvarani

Round 2
Reviewer 2 Report
This reviewer stands by the initial opinion for rejection.
Author Response
Dear Reviewer 2,
We appreciate your comment "This reviewer stands by the initial opinion for rejection.", but we also maintain our opinion that was given to you in the first round and that, unlike you, we authors, the other 3 reviewers and the magazine's editor believe in the potential of the article and that it be published, as the article has undergone a series of improvements, with contributions from other reviewers and editors. Unfortunately, we reiterate that we respect your opinion, but we do not agree with it, as the article does have publication potential.
Sincerely,
Felipe Masiero Salvarani
Reviewer 3 Report
Thank you for reviewing the English - it is still not perfect but should be able to be dealt with by the journal's copy editor.
A few more changes please:
Replace fore and hindlimbs with thoracic and pelvic limbs.
There is no ‘posterior’ aspect of animals, please use ‘caudal’.
p5: On what basis is the statement “The ruminal content of buffalo 01 and 05 was considered an unsuitable environment for bacterial activity” made? Please elaborate.
P5 What do you mean by “a fact that explains the triggering of thiamine”?
P5 What is meant by “the animals in this study were raised in a supportive system”?
See above
Author Response
Dear Reviewer 3,
We thank you again for your willingness to re-read our edited manuscript and for making new observations that have been corrected and clarified. Thank you very much for your attention and for supporting our work, making it even better.
- Corrections were made: fore and hindlimbs by thoracic and pelvic limbs; posterior’ aspect of animals, by ‘caudal’.
- On what basis is the statement “The ruminal content of buffalo 01 and 05 was considered an unsuitable environment for bacterial activity” made? The ruminal content found in of buffalo 01 and 05 was considered an unsuitable en-vironment for bacterial activity because contents of the rumen had a pasty appearance, with a “grease appearance” resulting from a mixture of mud and plant material (Figure 2A) and large number of abundant parasites (Figure 2B) also in the rumen. Therefore, as ruminants depend on the synthesis of thiamine by ruminal bacterial thiamine synthesis, it leads us to assume that in this ruminal unsuitable environment found in of buffalo 01 and 05, thiamine production was insufficient to meet the animal's' needs of the animal, not ruling out this condition in triggering the disease.
- What do you mean by “a fact that explains the triggering of thiamine”? The sentence really didn't make sense. What we wanted to say is that the rumen content found in buffaloes 01 and 05 was considered an unsuitable environment for bacterial activity and this fact may explain the triggering of PEM in these animals due to the non-synthesis of thiamine, which functions as the cofactor that regulates pentose. phosphate pathway responsible for ofmuch of the ATP used in nerve cells. (It has been corrected in the text).
- What is meant by “the animals in this study were raised in a supportive system”? This was a correction made by the MDPI professional, a specialist in the area, who we paid to review the quality of the English. But we agree that "supportive system" does not translate what we want to say in the article. Because the correct technical term is an extensive system, in which the animals are raised on pasture, free range, without supplementation of feed or mineral salt, feeding only on grass. (corrected in text).
Sincerely,
Felipe Masiero Salvarani